# Host Traits and Phylogeny Contribute to Shaping Coral-Bacterial Symbioses

Francesco Ricci,[a] Kshitij Tandon,[a,b] Jay R. Black,[c] Kim-Anh Lê Cao,[d] Linda L. Blackall,[a] Heroen Verbruggen[a]

aSchool of BioSciences, University of Melbourne, Victoria, Australia
bBiodiversity Research Center, Academia Sinica, Taipei, Taiwan
cSchool of Geography, Earth and Atmospheric Sciences, University of Melbourne, Victoria, Australia
dMelbourne Integrative Genomics, School of Mathematics and Statistics, University of Melbourne, Victoria, Australia

**ABSTRACT** The success of tropical scleractinian corals depends on their ability to establish symbioses with microbial partners. Host phylogeny and traits are known to shape the coral microbiome, but to what extent they affect its composition remains unclear. Here, by using 12 coral species representing the complex and robust clades, we explored the influence of host phylogeny, skeletal architecture, and reproductive mode on the microbiome composition, and further investigated the structure of the tissue and skeleton bacterial communities. Our results show that host phylogeny and traits explained 14% of the tissue and 13% of the skeletal microbiome composition, providing evidence that these predictors contributed to shaping the holobiont in terms of presence and relative abundance of bacterial symbionts. Based on our data, we conclude that host phylogeny affects the presence of specific microbial lineages, reproductive mode predictably influences the microbiome composition, and skeletal architecture works like a filter that affects bacterial relative abundance. We show that the $\beta$-diversity of coral tissue and skeleton microbiomes differed, but we found that a large overlapping fraction of bacterial sequences were recovered from both anatomical compartments, supporting the hypothesis that the skeleton can function as a microbial reservoir. Additionally, our analysis of the microbiome structure shows that 99.6% of tissue and 99.7% of skeletal amplicon sequence variants (ASVs) were not consistently present in at least 30% of the samples, suggesting that the coral tissue and skeleton are dominated by rare bacteria. Together, these results provide novel insights into the processes driving coral-bacterial symbioses, along with an improved understanding of the scleractinian microbiome.

**IMPORTANCE** The rapid decline of coral reefs, driven by climate changes, calls for manipulative interventions such as the use of probiotics, which can assist the resilience of these ecosystems. However, many knowledge gaps still exist in our understanding of coral-bacterial symbioses that need to be addressed before effectively applying interventions like probiotics. Here, by investigating the microbiomes of 12 common coral species we show that the associations with bacterial symbionts, thought to be critical to coral health, were influenced to some extent by host phylogeny, skeletal architecture, reproduction, and anatomical compartments. We therefore propose that fundamental and applied functional exploration of coral-associated microbes will help inform successful reef management measures.

**KEYWORDS** coral microbiome, symbiosis, bacteria, host traits, rare microbiome, coral microbiome

Address correspondence to Francesco Ricci, Riccifrancesco1989@gmail.com.

The authors declare no conflict of interest.

Ecology and evolution shape trait variation across species and populations, influencing host-microbiome associations (1, 2). Some of the interactions between the host and its associated microbial symbionts affect the fitness of the holobiont (3), leading to extraordinary evolutionary outcomes that have shaped life on Earth. Corals form a

holobiont with unicellular microalgae (Symbiodiniaceae) and a diverse range of microbes, which include bacteria, fungi, and viruses (4). The coral holobiont is considered to be an independent level of selection (5), but our understanding of the key mechanisms driving host-symbiont assemblages is limited. It has been observed that host evolutionary processes (6), skeletal architecture (7, 8), and mode of reproduction (9–12) contribute to microbiome composition; alongside these factors, a range of host ecological and morphological traits take part in the establishment and development of the coral microbiome.

The environment can be considered the main microbial reservoir for corals (9, 13, 14), and, for instance, the substrate underlying the colony is known to influence the holobiont structure (14). Host health status can be associated with shifts in the symbiotic microbial community that can disrupt the integrity of the holobiont (15, 16). The developmental stages of the host are related to successional processes of the microbiome that occur over time and influence the microbial richness (17). Disentangling the individual and combined effects of these factors is paramount to understanding microbial community assembly in corals.

Closely related species can harbor microbiomes that resemble each other, a pattern known as phylosymbiosis (18), which has been shown for several terrestrial and marine host-microbe systems (18, 19), including those of corals (6, 19). Phylosymbiosis can be driven by a range of mechanisms, including microbial filtering moderated by evolving host traits, ecological interactions with the host, or cophylogenetic relationship between the host and microbes (20, 21). However, the fraction of the coral microbiome influenced by these mechanisms remains unquantified.

Several coral traits can more directly affect the microbiome. Coral skeletal architecture differs among species (22), affecting the physicochemical properties of the colony (23) and the microbial biomass (7). For instance, light scattering is a function of skeletal density (8), resulting in different spatial gradients of light intensity and spectral composition that affect the microbiome (8, 24–26). Coral reproductive traits can influence the early microbiome, which plays a key role during coral ontogeny and affects the fitness of the host (10). Broadcast-spawning corals depend mostly on the acquisition of their symbionts from the environment (horizontal transmission) (9, 27, 28), as their gametes, embryos, and 3-day-old larvae are devoid of bacteria (29–31). Some brooding species exhibit a certain degree of vertical transmission (12, 27, 31), as bacteria have been found associated with the ectoderm layer in newly released planulae (12). However, whether coral bacterial communities can be predicted based on host traits is still to be investigated.

The coral microbiome is a dynamic system and provides an excellent case study of a highly diverse biological community contributing to holobiont success. Unravelling the composition of coral microbiomes is a necessary step toward understanding the interspecies relationships and functioning of the holobiont. To reduce the variability introduced by a range of factors known to affect the coral holobiont, such as colony age (17), spatial-temporal variability (32), and health status (33), we only sampled visibly healthy adult colonies from a small geographical location over 3 weeks. Using this approach, our study aimed to disentangle the influence of drivers affecting the microbiome composition, with a particular focus on host phylogeny, skeletal architecture, and reproductive mode. We also aimed to unravel the bacterial community structure and composition of the coral tissue and skeleton, including rare bacterial taxa.

## RESULTS AND DISCUSSION

**Experimental design and sequencing statistics.** We investigated and compared the structure and composition of the microbial communities of the coral tissue and skeleton, as well as those of surrounding seawater and sediment, and assessed differences among groups using multivariate analyses and ordination (Fig. 1). Then, to quantify the relative contribution of host phylogeny, skeletal architecture, and reproductive mode to the bacteria residing in the tissue and skeleton (Fig. 2), we performed

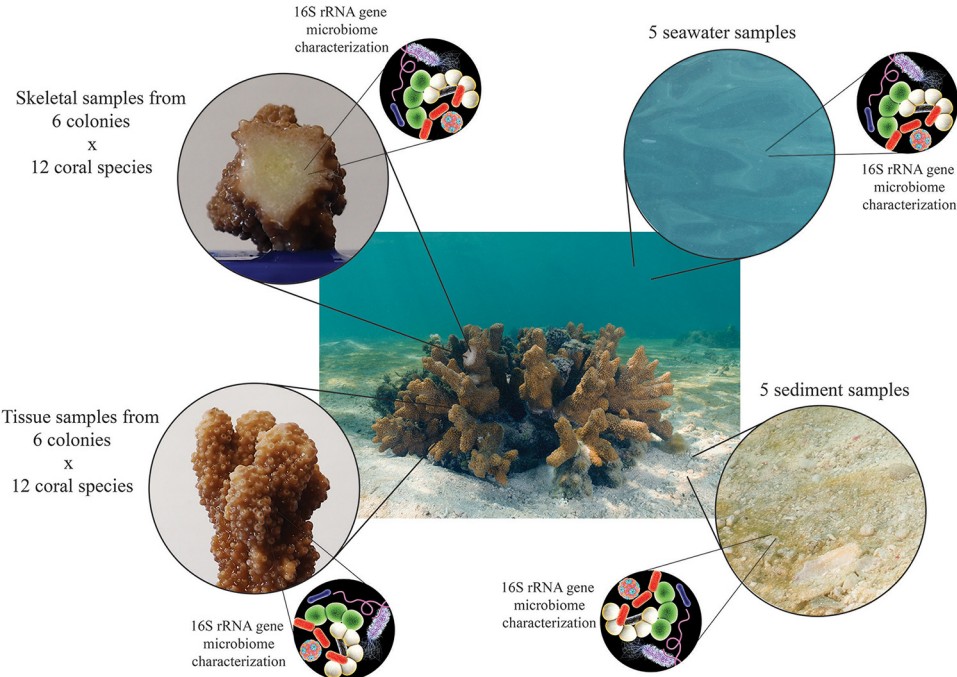

**FIG 1** Vignette of the experimental design aiming to answer the following questions: what are microbiome structures of the tissue and the skeleton of scleractinian corals, and how do they differ from the seawater and sediment microbiomes?

variation partitioning analysis. Finally, to identify the influence of these predictors on the presence and abundance of specific bacterial lineages, we used canonical correspondence analysis (CCA). Our sampling design also aimed to reduce factors known to affect the coral holobiont. Thus, we sampled only healthy adult colonies from a small geographic area, within 3 weeks, and we identified their tissue and skeletal prokaryotic communities using 16S rRNA gene amplicon sequencing because it provided the most comprehensive characterization of the bacterial community composition, including that of the rare bacterial taxa.

After denoising, amplicon sequence variant (ASV) filtering and contaminant removal, the 16S rRNA gene data set consisted of 6,977,071 reads (see Table S1 in the supplemental material) with an average length of 255 bp. The data set did not include unclassified reads (Table S1). The sediment samples contained 91,738 reads (minimum, 11,203; mean, 13,508; maximum, 17,475) resulting in 1,047 ASVs; the seawater samples contained 67,540 reads (minimum, 15,946; mean, 18,348; maximum, 23,430) resulting in 1,026 ASVs; and the coral samples contained 6,817,793 reads (minimum 1,993; mean, 47,872; maximum, 115,669) resulting in 14,910 ASVs.

**Coral tissue and skeletal microbiomes overlap but differ in their relative abundances.** By comparing the coral tissue and skeleton microbiome (Fig. 1) our work shows that a large fraction of bacteria can colonize both anatomical compartments (i.e., 86% of tissue ASVs were found in at least one skeleton sample, and 56% of skeletal ASVs were found in at least one tissue sample). Thus, our results support the hypothesis formulated by Marcelino et al. (34) that the skeleton can serve as a reservoir for coral tissue microbes. On the basis of this, it can be hypothesized that after a period of dysbiosis, beneficial bacteria could quickly repopulate the tissue from the skeleton (34). Despite this, the $\beta$-diversity of coral tissue and skeletal microbiomes differed significantly in 8 out of 12 coral species (P values in the range of 0.001 to 0.016; see Fig. S1), with the exceptions of *Isopora palifera* (P = 0.053), *Montipora digitata* (P = 0.075), *Goniopora tenuidens* (P = 0.061), and *Platygyra daedalea* (P = 0.21). Therefore, our results indicate that while many bacteria are not selective in colonizing the coral tissue or skeleton, the complex

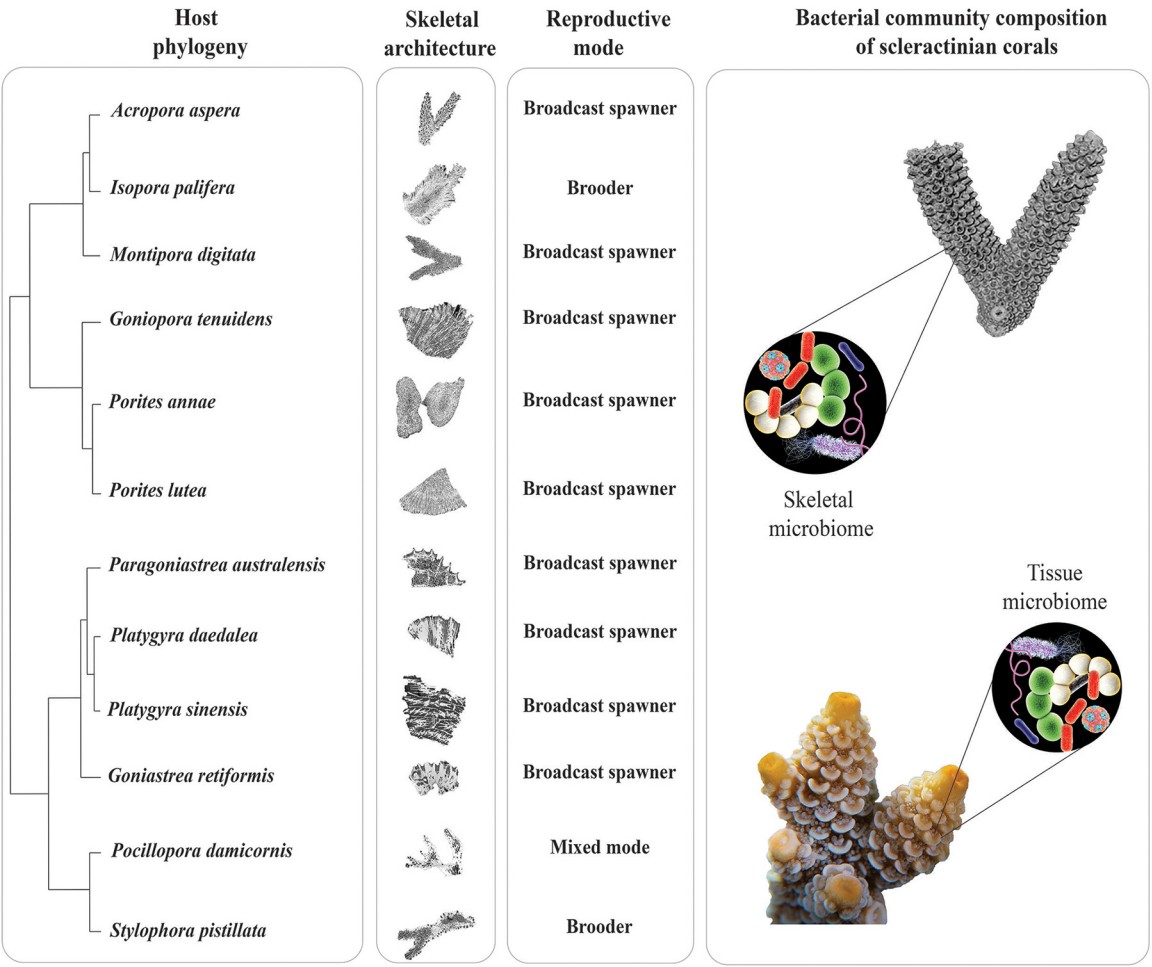

**FIG 2** Vignette of the experimental design aiming to answer the question: to what extent host phylogeny, skeletal architecture and reproductive mode affect the microbiome composition of tissue and skeleton of scleractinian corals?

array of biotic and abiotic interactions characteristic of each compartment shapes the microbiome composition differently (Fig. S1).

Coral tissue and skeleton are also known to differ in their physicochemical environment, with the skeleton offering a wide array of microniches (23), and one study reported that the skeletal communities are more diverse than their tissue counterparts (6). Our data showed comparable α-diversities of tissue and skeleton microbiomes for most coral species (Fig. S2). This conflicts with the findings of Pollock et al. (6), who found differences in the α-diversities of the two anatomical compartments; while it may represent an underlying biological cause, methodological differences between the studies may also contribute to the observed differences (e.g., using ASVs versus operational taxonomic units [OTUs] for taxonomic resolution).

**Host traits and phylogeny shape the bacterial community composition.** We quantified the influence of some of the most significant processes known to influence the establishment and development of the coral microbiome (Fig. 2), such as host phylogeny (6), skeletal architecture (7), and reproductive mode (9, 10), by applying variation partitioning analysis to the tissue and skeletal bacterial communities. Our models explained 14% of the tissue and 13% of the skeletal microbiome variation, leaving a high proportion of the variation (>85%) unexplained by the predictors (Fig. 3a and d). This is not entirely surprising, since past reports have found that the coral microbiome is variable within a single coral colony (35) and among host genotypes (36), species, and reef habitats (37–39). We hypothesize that a combination of unmeasured

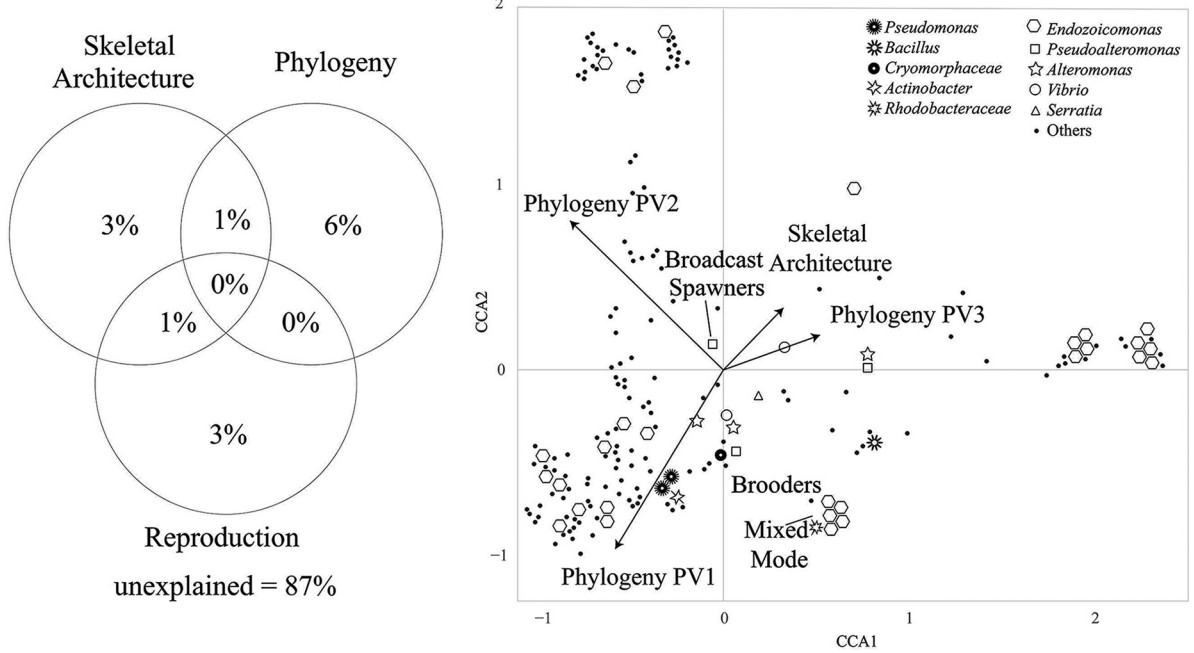

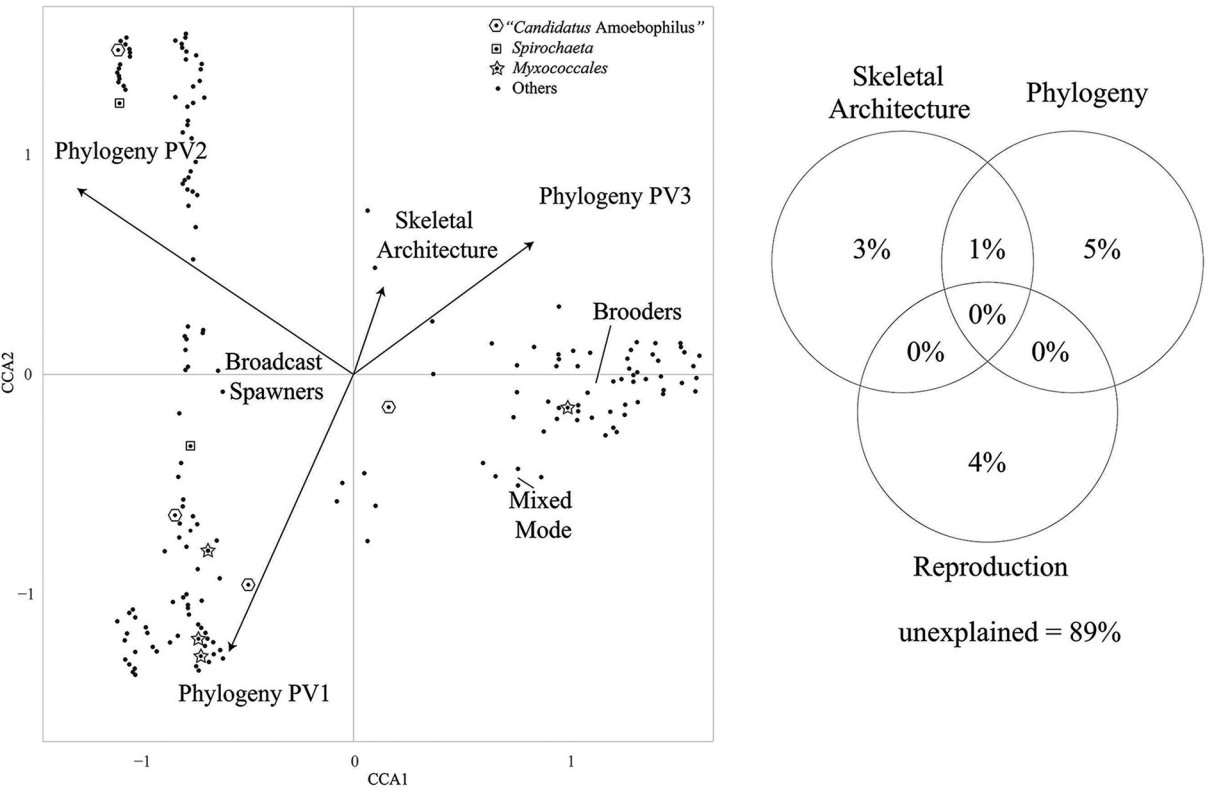

**FIG 3** Partitioning of variation in the bacterial community composition of coral tissue (a) and skeleton (d) explained by host skeletal architecture, phylogeny, and reproductive mode. Adjusted $R^2$ values are given. Canonical correspondence analysis (CCA) biplots represent the tissue (b) and skeletal (c) microbiome structures according to each explanatory variable. Arrows represent the quantitative explanatory variables skeletal architecture and phylogeny (PV1, PV2, and PV3), with arrowheads indicating the direction of increase. The categorical explanatory variables broadcast spawners, brooders, and mixed mode are positioned on the biplot according to their coordinates. All of the explanatory variables in both CCA biplots were highly significant ($P < 0.001$). In variation partitioning analysis (a, d), the total explained and unexplained variance can exceed 100%.

environmental variables, functional redundancy (same functions putatively conveyed by different bacterial taxa), community assembly processes, and physicochemical properties specific to each coral colony take part in determining the coral holobiont assemblage and limiting its broader consistency.

**Host phylogeny drives key coral-bacterial associations.** Employing linear discriminant analysis effect size (LeFSe) analysis, we identified members of the coral microbiome that were present at higher abundance in the coral tissue or skeleton (Fig. 4 and Table S2); then, through CCA analysis, we assessed their association with host phylogeny and coral traits. In the CCA analysis, the three coral phylogenetic variables (PV1, PV2, and PV3; see "Host phylogeny" in Materials and Methods) showed a strong association with some ASVs (Fig. 2b and c).

*Endozoicomonas* species were found at higher relative abundance in the coral tissue (33.30% of the tissue ASVs and 7.43% of the skeleton ASVs across the whole data set; see Fig. 4 and Table S2), and individual ASVs belonging to this genus were more abundant in corals of the robust clade (Fig. 3b; positive correlation with PV1) and the families Poritidae (Fig. 3b; positive correlation with PV2) and Pocilloporidae (Fig. 3b; positive correlation with PV3). Our results confirmed *Endozoicomonas* being a highly prevalent genus of coral symbionts across coral species (40, 41); given their putative beneficial roles, which include nutrient provision and host homeostasis support (41, 42), it seems possible that these bacteria developed a mutualistic symbiosis with corals over time.

*Alteromonas* and *Pseudoalteromonas* species were found at higher relative abundances in the tissue (Fig. 4 and Table S2) of robust corals (0.74% and 0.37% of the tissue ASVs and 0.05% and 0.08% of the skeleton ASVs; Fig. 3b) and in the family Pocilloporidae (0.14% and 0.25% of the tissue ASVs and 0.05% and 0.08% of the skeleton ASVs; Fig. 3b). Additionally, *Pseudoalteromonas* species were associated with the family Poritidae (Fig. 3b). These bacteria are thought to take part in nitrogen cycling and antibacterial activity (43, 44), and because of these functional features, they could play pivotal roles in the holobiont of these coral lineages. We hypothesize that the absence of correlation between *Alteromonas* species and Poritidae, which suggests a low relative abundance of these bacteria in this coral lineage, could be driven by competition for similar resources between these and other bacteria.

Bacteria of the genera *Spirochaeta* and "*Candidatus* Amoebophilus" were found at higher relative abundances in the coral skeleton (1.29% and 14.60% of the skeleton ASVs and 0.19% and 0.97% of the tissue ASVs; Fig. 4 and Table S2), and individual ASVs belonging to these genera correlated with corals in the robust clade and in the family Poritidae (Fig. 3c). *Spirochaeta* species usually thrive in oxygen-deprived environments (45) such as the coral skeleton and, given their ability to fix nitrogen and carbon (46), could be key members of the community of the skeletal environment. Despite "*Candidatus* Amoebophilus" having been flagged as a member of the coral core microbiome (24, 38), its role in the holobiont is still unclear. Available data on "*Candidatus* Amoebophilus" (47) and "*Candidatus* Amoebophilus asiaticus" (48) show reduced genomes with limited metabolic capabilities, suggesting they may rely on the host for survival. Interestingly, the "*Candidatus* Amoebophilus asiaticus" genome harbors a high count of eukaryotic domain-like proteins, which include Ankyrin repeats and WD40 repeat domain proteins. These proteins are used by intracellular pathogens, as well as by symbiotic bacteria, to interact with hosts and modulate host response via a multitude of protein-protein interactions (48). A wide arsenal of eukaryote-like proteins has been reported in many coral-associated bacterial groups (49).

Bacteria in the order *Myxococcales* were found at higher relative abundance in the coral skeleton (5.82% of the skeleton ASVs and 0.63% of the tissue ASVs across the whole data set; Fig. 4 and Table S2) and were correlated with corals in the robust clade (Fig. 3c). Moreover, recent studies have shown that these microbes might have codiversified with corals (6). *Myxococcales* species are known to play beneficial roles in other systems, such as agricultural settings (50), where they keep pathogen populations under control by releasing large quantities of antibiotics. The skeletons of many

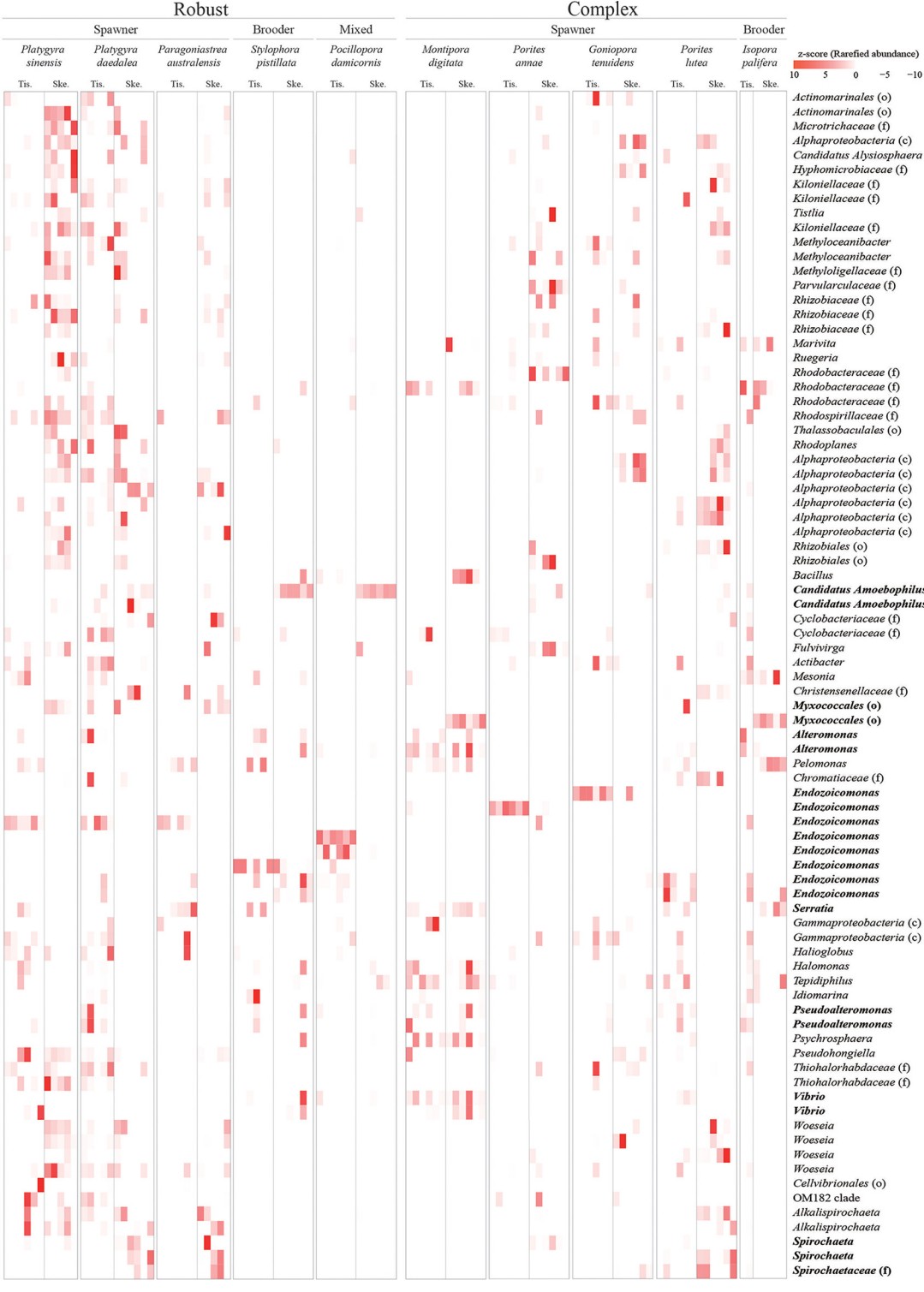

**FIG 4** Heatmap resulting from the linear discriminant analysis effect size (LEfSe) analysis showing differentially abundant bacterial amplicon sequence variants (ASVs) between coral tissue and skeleton. Specifically, ASVs with a log(LDA) of >3 (Kruskal-Wallis test: $P < 0.05$) and present in at least 10 samples (inclusive of tissue and skeleton) were considered differentially abundant, and their abundance was Z-score transformed (legend on the top right; darker red indicates more abundant bacteria). On the top are reported the tissue (Tis.) and skeleton (Ske.) of each coral species analyzed in this study, their clade (robust or complex), and their reproductive mode (spawner, brooder, or mixed). On the left are reported the taxonomic classification of the differentially abundant bacterial ASVs at the genus, family (f), order (o), or class (c) level. Taxa indicated in boldface represent the bacterial groups discussed in this study. Tissue and skeleton microbiomes differed in terms of bacterial abundances. Bacteria that preferentially colonized the tissue included *Alteromonas, Endozoicomonas, Halomonas, Pseudoalteromonas, Vibrio,* and *Woesia,* while among those that preferentially colonized the skeleton were "*Candidatus* Amoebophilus," *Kiloniellaceae, Myxococcales, Rhizobiales,* and *Spirochaeta.* The heatmap also shows that the microbiome of each coral species is characterized by a few abundant and many rare bacterial taxa. The coral species *A. aspera* and *G. retiformis* are not shown on the heatmap because LeFSe analysis did not detect differences between their respective tissue and skeletal microbiomes.

corals analyzed in this study concurrently harbored a high relative abundance of sequences affiliated with *Myxococcales* and potentially pathogenic bacteria (i.e., *Vibrio* and *Serratia*). Thus, as reported for agricultural settings, and given that all the sampled colonies were visibly healthy, *Myxococcales* species may play similar roles in the coral holobiont by controlling pathogen populations (51).

**Skeletal architecture affects microbial abundance.** Micro-computed tomography (micro-CT) analysis allowed us to characterize the skeletal architecture of each coral species and showed the variation in porosity within and across coral species (Fig. 5g). The characteristic skeletal architecture of each coral species explained a portion of the tissue and skeletal microbiomes variations (Fig. 3a and d). The skeleton affects the physicochemical properties of the whole coral colony, including the tissue, and Marcelino et al. (8) found that, by refracting light back, the skeleton transports and redistributes light across the colony and alters the light environment in the tissue, affecting the Symbiodiniaceae. One could propose that this also causes downstream effects on other members of the tissue microbiome. Accordingly, we found that skeletal architecture was associated with several key members of the tissue microbiome, including *Endozoicomonas*, *Alteromonas*, and *Pseudoalteromonas* (Fig. 3b). Our results also suggest that skeletal architecture could alter the relative abundance of bacterial species rather than filtering them out entirely. For instance, we found that despite the bacterial genera *Bacillus*, *Halomonas*, and *Vibrio* being mainly associated with coral skeletons with larger pore sizes, such as *M. digitata* and *Acropora aspera* (Fig. 5g and Table S3), they were also present in species with more dense skeletons at low relative abundances (e.g., *Pocillopora damicornis*, *Isopora palifera*, and *Stylophora pistillata*; see Fig. 5g and Table S3). Similarly, the bacterial genera *Pseudohaliea* and *Rhodoplanes*, which we found at higher relative abundances in species with more dense skeletons (e.g., *Goniastrea retiformis*, *I. palifera*, and *Porites lutea*; see Fig. 5g and Table S3), were also present in coral skeletons with larger pore sizes (e.g., those of *Platygyra sinensis* and *M. digitata*; see Fig. 5g and Table S3) at low relative abundances. While our data do not allow us to disentangle the causative factors determining the differences in relative abundance of bacteria across coral species with porous or dense skeletons, it is possible that the skeletal architecture could affect the microniches of the coral colony and determine a more suitable or hostile environment for specific bacteria.

**Reproductive mode influences the microbiome composition in a predictable manner.** Variation partitioning analysis showed that reproductive mode explained a portion of the microbiome variation (tissue, 3%; skeleton, 4%; see Fig. 3a and d). Additionally, CCA analysis revealed that ASVs associated with the bacterial taxa *Acinetobacter* spp., *Bacillus* spp., *Cryomorphaceae*, *Endozoicomonadaceae*, *Pseudomonas* spp., and *Rhodobacteraceae* correlated with the reproductive mode variables (Fig. 3b). These bacterial taxa were reported in studies whose focus was the coral microbiome establishment (9–11, 52), and some of them are thought to play important roles, including in nutrient provision and support of host homeostasis (41–44). Although our analysis was not conceived to investigate the microbiome establishment, the correspondence between our findings and those of past reports suggests that coral reproductive mode could predictably influence the microbiome composition and that some early host-symbiont associations may persist across a coral's lifetime.

Our data show that tissue and skeletal bacteria correlated with the reproductive mode variables (broadcast spawners, brooders, and mixed mode; see Fig. 3b and c) and were associated with key holobiont members, including *Endozoicomonas*, *Alteromonas*, *Pseudoalteromonas*, and *Myxococcales* (Fig. 3b and c). These bacteria are all known for their putative beneficial roles and could help the host by being involved in processes such as nutrient cycling and support of homeostasis (42–44, 51), which could also facilitate the early developmental stages of the coral. In the tissue, the reproductive mode was also associated with bacterial taxa such as *Serratia* and *Vibrio* (Fig. 2b), which are known as potential pathogens in some coral species (11, 53, 54). As suggested in previous studies, under normal conditions, these putative pathogens are commensal members of the holobiont, while their detrimental potential could emerge during dysbiotic states (55).

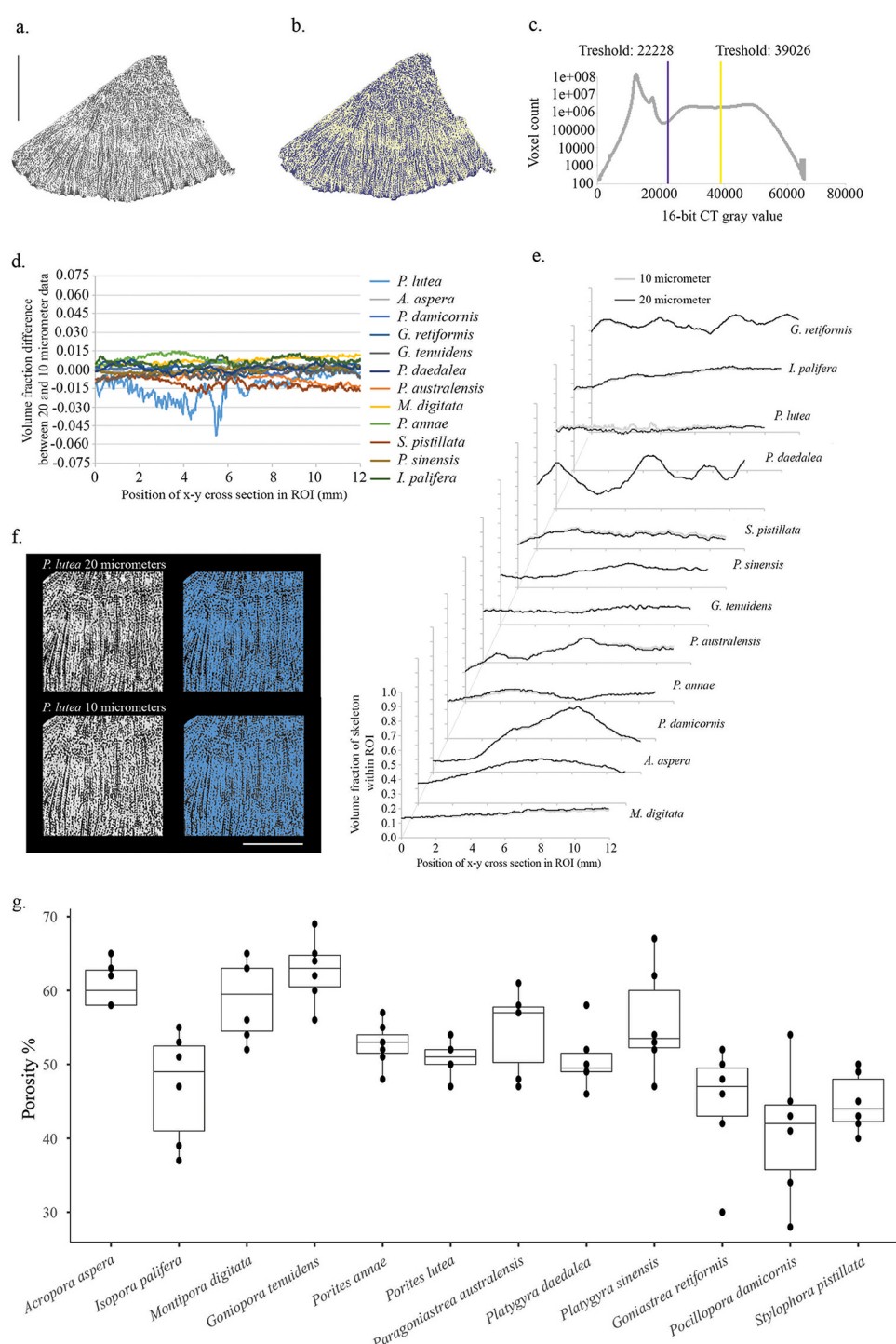

**FIG 5** (a) Representative slice of micro-computed tomography (micro-CT) data through a sample of *Porites lutea* (FRH49) showing denser structure as brighter gray to white values and less dense structure, such as air, as dark gray to black values surrounding the specimen. Black scale bar, 10 mm. (b) The same slice segmented into three phases, as follows: skeletal material (shaded yellow); organic matrix (shaded purple); and air/bubble wrap within and surrounding the specimen (unshaded black). (c) Grayscale histogram of 16-bit micro-CT data showing the points of segmentation between the different phases. The two peaks within the lowest density phase represent the air within and surrounding the specimen (peak around 12,000) and the bubble wrap and plastic specimen holder (peak around 17,000). (d) Difference between the 20-$\mu$m and 10-$\mu$m data trends highlighting the very small variability within the porosity trends, with no consistent trend toward an under- or overestimate of porosity at one given resolution. (e) Plots of skeletal volume trends within the *x*-*y* cross-section of a 12 mm by 12 mm by 12 mm region of interest (ROI) as a function of distance along the *z* axis of the ROI for both 20-$\mu$m and 10-$\mu$m micro-CT data. (f) Representative slice (at 4.3 mm) from a *P. lutea* sample comparing the 20-$\mu$m to 10-$\mu$m micro-CT data (left subpanels) and segmented skeleton in both (right subpanels). White scale bar, 6 mm. (g) Box plots showing the variability of porosity within and across coral species.

**Coral microbiomes are dominated by rare bacterial species.** Except for a few bacteria that were consistently present across several coral species (e.g., *Endozoicomonas*, "*Candidatus* Amoebophilus," *Fulvivirga*, *Vibrio*, *Alteromonas*, and *Pseudoalteromonas*), ASVs affiliated with rare bacterial taxa dominated the coral microbiomes. The great majority of bacterial ASVs (99.6% for tissue and 99.7% for the skeleton) were not consistently present in at least 30% of the samples of our data set. The core microbiome, i.e., the ASVs occurring in at least 30% of samples, consisted of only 16 ASVs in the tissue (12 *Gammaproteobacteria* ASVs, 1 *Alphaproteobacteria* ASV, 1 *Deltaproteobacteria* ASV, 1 *Bacilli* ASV, and 1 *Bacteroidia* ASV; see Table S4) and 9 in the skeleton (2 *Gammaproteobacteria* ASVs, 2 *Alphaproteobacteria* ASVs, 2 *Bacteroidia* ASVs, 2 *Bacilli* ASVs, and 1 *Spirochaeta* ASV; Table S4). Even fewer ASVs were present when we analyzed the core microbiome at higher thresholds, i.e., 5 ASVs occurring in at least 50% of tissue samples (4 *Gammaproteobacteria* ASVs and 1 *Bacilli* ASV) and 1 single ASV in the skeleton (*Bacteroidia*; see Table S4). Finally, only 1 ASV in the tissue (*Gammaproteobacteria*) and 1 in the skeleton (*Bacteroidia*; see Table S4) occurred in at least 60% of samples, and none were present at higher thresholds. The prevalence of few abundant and many rare bacterial lineages is clearly apparent when the composition of the coral microbiomes is displayed as a heatmap (Fig. 4).

A closer look at microbiome variability within species showed that even within closely related hosts, most bacterial ASVs were not present across individuals of the same species (Table S5), leading to wide variability between samples. The skeletal microbiome showed a higher proportion of core members than the tissue microbiome in 9 out of 12 coral species (Table 1 and Table S5). Since the microbial communities of tissue and skeleton were dominated by rare bacterial ASVs (Table 1 and Table S5), we analyzed their Pielou's evenness, a measure indicating how similar the abundances of different species in the microbiome are (56). Despite finding comparable evenness of the tissue and skeletal bacterial communities for all of the species confounded (Fig. S3a), when we compared across coral species, we found high variability in the evenness of both tissue and skeletal microbiome (Fig. S3b and S3c). For instance, in *G. retiformis*, both tissue (Pielou's evenness mean, 0.87) and skeletal (Pielou's evenness mean, 0.74) microbiomes showed evenly distributed bacterial populations, while in *S. pistillata*, the tissue microbiome (Pielou's evenness mean, 0.39) showed variability across individuals, and the genera *Endozoicomonas* and "*Candidatus* Amoebophilus" dominated the skeletal microbiome (Pielou's evenness mean, 0.20). In line with our results, it has been previously reported that the microbiome composition of each coral species can show various degrees of diversity (57), and in some cases, one or a few bacterial taxa can be dominant, like in *Pocillopora verrucosa*, whose bacterial community is dominated by the genus *Endozoicomonas* (58). A range of processes that includes host evolution (6), traits (7, 8, 10), microniche partitioning (23), priority effects, and functional redundancy (59, 60) may synergistically affect the microbiome assembly and ultimately determine the variability of corals' bacterial communities.

**Seawater and sediment as potential microbial reservoirs for the coral microbiome.** We compared the coral tissue and skeleton microbiomes with those of seawater and sediment (Fig. 1). Although our results suggest that the $\beta$-diversity of coral tissue and skeletal microbiomes differed from those of seawater and sediment (Fig. S4), a component of the tissue and skeletal microbiome overlapped with that of the environment (Table 2). Our comparison of the coral microbiota with those of the surrounding seawater and sediment showed that bacterial species shared between corals and their environment at the time of sampling accounted for more than 30% of the ASVs in *A. aspera* and *P. damicornis*, and the values for the other species were in the range of 10.5 to 27.5% (Table 2). Processes such as microbial dispersal across space and environmental heterogeneity can influence the structure of host-microbial systems (61), including those of corals (32). Accordingly, our results suggest that the bacteria present in the surrounding environment may function as potential environmental reservoirs for corals and, given that our seawater and sediment samples were taken over a period of

**TABLE 1** Percentages of rare and core ASVs present in the tissue and skeleton of each coral species

| Coral species and compartment | Rare ASVs (%) | Core ASVs (%) |
|---|---|---|
| *Porites lutea* tissue | 94.7 | 5.3 |
| *Porites lutea* skeleton | 72.2 | 27.8 |
| *Paragoniastrea australensis* tissue | 68.5 | 9.2 |
| Paragoniastrea australensis skeleton | 71.6 | 28.4 |
| *Acropora aspera* tissue | 77.2 | 22.8 |
| *Acropora aspera* skeleton | 72.6 | 27.4 |
| *Montipora digitata* tissue | 80.0 | 20.0 |
| *Montipora digitata* skeleton | 79.9 | 20.1 |
| *Pocillopora damicornis* tissue | 80.8 | 19.2 |
| *Pocillopora damicornis* skeleton | 83.1 | 16.9 |
| *Porites annae* tissue | 74.3 | 25.7 |
| *Porites annae* skeleton | 82.2 | 17.8 |
| *Goniastrea retiformis* tissue | 80.7 | 19.3 |
| *Goniastrea retiformis* skeleton | 71.2 | 28.8 |
| *Stylophora pistillata* tissue | 82.4 | 17.6 |
| *Stylophora pistillata* skeleton | 79.3 | 20.7 |
| *Goniopora tenuidens* tissue | 84.4 | 15.6 |
| *Goniopora tenuidens* skeleton | 79.0 | 21.0 |
| *Platygyra sinensis* tissue | 85.8 | 14.2 |
| *Platygyra sinensis* skeleton | 65.5 | 34.5 |
| *Platygyra daedalea* tissue | 80.1 | 19.9 |
| *Platygyra daedalea* skeleton | 81.5 | 18.5 |
| *Isopora palifera* tissue | 91.5 | 8.5 |
| *Isopora palifera* skeleton | 83.8 | 16.2 |

1 month, it seems likely that a more prolonged sampling would recover a higher fraction of the coral microbiome. The seasonally variable microbiome present in the environment (62, 63) can offer a large pool of important bacterial symbionts to be sourced by the coral holobiont through processes like selection and winnowing (9, 64).

**Unusual suspects show persistent associations with corals.** Our work identified several bacteria that were consistently associated with corals but are either new to the coral microbiome field or understudied. *Cyclobacteriaceae* taxa accounted for 7.30% of the ASVs of the data set and were present in several samples of every coral species (Table S6). Currently, *Cyclobacteriaceae* have only been reported in fire coral colonies (36), in 7-month-old *Acropora* recruits (65), and in intracolony changes in abundance after bleaching of *Acropora* spp. (66). Our work shows that this family is much more widespread across the complex and robust clades. From a functional point of view, members of this family could benefit the coral through carbohydrate metabolism, carotenoid biosynthesis, antibiotic resistance, and quorum-sensing regulation (67).

Bacteria in the genus *Paramaledivibacter* accounted for 2.92% of the data set ASVs and were found in multiple coral species (Table S6). This genus of strictly anaerobic bacteria was only reported recently in the coral literature (68). Information about these bacteria is still scarce, but given their ability to degrade amino acids and peptides, they could exploit the resources of the host (69). As mentioned above, we hypothesize that detrimental effects by pathogens could be counterbalanced by beneficial holobiont members, including *Myxococcales* and *Pseudoalteromonas*. Despite *Paramaledivibacter* being described as strictly anaerobic, we did find these bacteria in the coral tissue. This finding may seem unusual, but this is not the first study reporting obligate anaerobes in coral compartments known to be oxic (70, 71).

*Roseospira* has also not been previously reported in the coral literature, but ASVs affiliated with these bacteria accounted for 0.29% of the data set and were found in the species *P. sinensis*, *P. daedalea*, *P. lutea*, *Paragoniastrea australensis*, *G. retiformis*, and *Goniopora tenuidens* (Table S6). These purple nonsulfur bacteria seem to be able to colonize a diverse range of environments and grow optimally under

**TABLE 2** Percentages of ASVs retrieved from seawater and sediment that were concurrently present in each coral species

| Coral species | Seawater ASVs (%) | Sediment ASVs (%) |
|---|---|---|
| *Acropora aspera* | 19.5 | 11.9 |
| *Goniastrea retiformis* | 6.1 | 8.4 |
| *Goniopora tenuidens* | 8.2 | 9.6 |
| *Isopora palifera* | 14.4 | 10.9 |
| *Montipora digitata* | 15.7 | 10.3 |
| *Paragoniastrea australensis* | 4.3 | 6.2 |
| *Platygyra daedalea* | 5.5 | 8.3 |
| *Platygyra sinensis* | 4.3 | 6.2 |
| *Pocillopora damicornis* | 19.1 | 13.7 |
| *Porites annae* | 12.1 | 9.8 |
| *Porites lutea* | 8.1 | 10.5 |
| *Stylophora pistillata* | 17.0 | 10.5 |

photoheterotrophic conditions (72). Thus, given their ability to utilize substrates known to be present in corals, such as acetate (73) and glutamate (74), and to use near-infra-red wavelengths not absorbed by Symbiodiniaceae species, the coral colony could offer an array of microenvironments where these bacterium's niche preferences are met.

**Conclusions.** Through a combination of a homogeneous experimental design that minimizes external biases affecting the microbiome; the use of innovative technologies, including micro-CT scanning to evaluate host traits; and the application of a range of statistical analyses, our study allowed us to unravel the structure of the coral microbiome and quantify how it is influenced by host phylogeny, skeletal architecture, and reproductive mode.

We show that host phylogeny and traits explained 14% of the tissue and 13% of the skeletal microbiome composition and were associated with a range of microbial partners thought to affect holobiont health and functioning. Based on our results, we hypothesize that reproductive mode may influence the microbiome composition in a predictable manner, while skeletal architecture works like a filter affecting bacterial relative abundance. Although our analysis accounted for some of the most influential processes known to affect the microbiome composition, these could only marginally explain the microbiome variation of tissue and skeleton. A holistic view of the mechanisms determining the holobiont composition will be gained by assessing the influence of the physicochemical and dynamic biochemical environment of the coral colony on the microbiome composition and by assessing whether the presence of some bacteria (whether dominant or rare) may affect the overall structure of the microbiome. In this study, we provided substantial evidence that coral tissue and skeleton microbiomes are dominated by rare taxa and differ in bacterial abundance, but a consortium of bacteria can colonize both compartments and the skeleton can function as a microbial reservoir.

While our study answers several unsolved questions about the bacterial community structure of scleractinian corals and the mechanisms driving its composition, it also exposes knowledge gaps. Despite our focus on commonly studied coral species, we identified three bacterial groups that are understudied or were not previously reported in the coral literature. This highlights that a full characterization of the taxonomic composition of the coral microbiome has still not been achieved. Substantial further work will be needed to fully understand its functions in the coral holobiont, its fine-scale distribution in relation to ecological microniches, and the metabolic hand-offs that happen among microbiome members and with the host. The use of putative beneficial microorganisms has been proposed as a tool to mitigate the increasing pressure of anthropogenic activities on coral reefs (75); therefore, we hope that the detailed knowledge gained from our study about persistent association between specific coral

lineages and bacterial taxa can form the basis for further advances in probiotic strategies to improve coral resilience in future climate scenarios.

## MATERIALS AND METHODS

**Sample collection and processing.** To adhere to sampling limits advised by the Great Barrier Reef (GBR) Marine Park Authority and to contain the impact of this project on the natural environment, we collected a total of 72 (6 each of 12 coral species) coral fragments (collected at low tide in water 0 to 1 m deep, depending on tides) belonging to the robust (*Goniastrea retiformis*, *Paragoniastrea australensis*, *Platygyra daedalea*, *Platygyra sinensis*, *Pocillopora damicornis*, and *Stylophora pistillata*) and complex clades (*Acropora aspera*, *Goniopora tenuidens*, *Isopora palifera*, *Montipora digitata*, *Porites annae*, and *Porites lutea*), five seawater samples (5 L each) and five 50-mL sediment samples from the research zone of Heron Island reef flat (maximum distance between samples, ~300 m), central Great Barrier Reef (23°44′S, 151°91′E), during January 2020. Corals were collected using a set of sterile hammers and chisels and, along with seawater, placed in zipper lock polyethylene bags sterilized before use by bleach and ethanol wash. Coral tissue was removed from the skeleton of each sample using a Waterpik and sterile seawater (SSW), and then tissue slurry and skeletal fragments were collected and snap-frozen by immersion in liquid nitrogen and stored at −80°C until processing. The seawater samples and 2 samples of SSW (5 L each) were filtered using 0.22-$\mu$m filters (MilliporeSigma) before snap-freezing. Because of the comparative nature of this study, we took a range of precautions to avoid any potential cross contamination and computationally removed potential contaminants. For instance, we used sterile tweezers and razor blades for every specimen, we sampled skeletal fragments 5 mm from the tissue to prevent tissue-skeleton microbiomes cross contaminations and sequenced SSW used to remove the coral tissue and control samples taken during the DNA extraction and amplification. Sequences associated with potential contaminants retrieved from SSW and control samples were computationally removed using the R package *decontam* (76).

**Library preparation, sequencing, and initial quality control.** The total DNA of each coral tissue, coral skeleton, seawater, sediment, or control sample was extracted using the Wizard Genomic DNA purification kit (Promega). Extractions were also performed on eight blanks (used to account for kit contamination) taken during both the extraction and amplification protocols. SSW and blanks served as controls. We used a two-step PCR (PCR) amplification, with the first amplifying the target marker and the second adding Illumina adapters (underlined in the following sequences). The V5-V6 regions of the 16S rRNA gene were PCR amplified using the following primer pairs: 784F (5′-<u>TCGTCGGCAGCGTCAGATGTGTATAAGAGACAG</u> AGGATTAGATACCCTGGTA-3′) and 1061R (5′-<u>GTCTCGTGGGCTCGGAGATGTGTATAAGAGACAG</u>CRRCACGAG CTGACGAC-3′) (76).

The first PCR round was conducted in 20-$\mu$L reaction mixtures using the Kapa HiFi HotStart ReadyMix and 10 $\mu$L of l M each primer (working concentration), with a thermal cycling profile of 95°C for 3 min; 25 cycles of 98°C for 20 s, 60°C for 15 s, and 72°C for 30 s; and a final extension at 72°C for 1 min. The second PCR round was conducted in 20-$\mu$L reaction mixtures using the GoTaq Green mix and 10 $\mu$M each custom-made Illumina index. The thermal cycling profile was as follows: 95°C for 3 min; 24 cycles each at 95°C for 15 s, 60°C for 30 s, and 72°C for 30 s; and a final extension at 72°C for 7 min. Samples, blanks, and controls were sequenced on the Illumina MiSeq platform (2 × 300-bp paired end reads) at the Walter and Eliza Hall Institute of Medical Research. We note that the total number of cycles ($N = 49$) in the 2-step PCR amplification could have led to an accumulation of nonspecific products; however, the quality of the PCR products was tested on a TapeStation (model 4200) at the Walter and Eliza Hall Institute of Medical Research. Sequences were processed using the QIIME 2 pipeline version 2020.11 (77). Cutadapt was used to remove primers (78). DADA2 was used to merge forward and reverse reads, remove poor-quality sequences, perform dereplication, and eliminate chimeras (79). Taxonomy was assigned using the feature classifier plugin built into the QIIME 2–SILVA v132 QIIME release (80).

**Host phylogeny.** By using skeletal morphology and structures information retrieved from Corals of the World (http://www.coralsoftheworld.org/page/home/) (81) we confirmed the identity of the targeted corals to the species level. The phylogeny of the 12 coral species was extracted from the multigene molecular phylogeny of corals published by Huang and Roy (82). We took their set of supertrees and derived a consensus tree in TreeAnnotator (BEAST 2.4.8) (83). We computed a matrix of pairwise patristic distances from the phylogenetic tree with the "distTips" function from *adePhylo* (84) and performed nonmetric multidimensional scaling (NMDS; isoMDS from *MASS*) (85) to obtain a set of variables that can be used to relate the host phylogeny to the microbiome in multivariate analyses. The vectors of the positions of the coral species along the three dimensions of the NMDS were used to represent host phylogeny in downstream analyses, dividing it into three main phylogenetic variables (PVs). PV1 separated the robust from the complex corals, correlating positively with the robust clade (*G. retiformis*, *P. australensis*, *P. daedalea*, *P sinensis*, *P damicornis*, and *S. pistillata*) and negatively with the complex clade (*A. aspera*, *G. tenuidens*, *I. palifera*, *M. digitata*, *P. annae*, and *P. lutea*). PV2 represents the phylogenetic subdivision of complex species into the Poritidae (*G. tenuidens*, *P. lutea*, and *P. annae*), which correlated positively with PV2, and the Acroporidae (*A. aspera*, *I. palifera*, and *M. digitata*), which correlated negatively with PV2. PV3 represents the phylogenetic subdivision of the robust species into the Pocilloporidae (*P. damicornis* and *S. pistillata*), which correlated positively with PV3, and the Merulinidae (*G. retiformis*, *P. australensis*, *P. daedalea*, and *P. sinensis*), which correlated negatively with PV3.

**Skeletal architecture.** Portions of all 72 coral fragments were scanned using a Phoenix Nanotom M micro-CT scanner (Waygate Technologies), capturing the full specimen structure at a voxel resolution of 20 $\mu$m. Further to this, a 12 mm by12 mm by 12 mm region of interest (ROI) from one sample from each

of the 12 species was scanned at a 10-$\mu$m resolution. Preserved samples were scanned in air and secured within a specimen jar with bubble wrap to prevent movement during scanning. Scans were collected using the datos|x acquisition software (Waygate Technologies) with an X-ray energy of 110 kV and 300 mA with a tungsten target and a 0.1-mm copper filter to preharden the X-ray beam. A fast-scan setting was used to collect between 1,199 and 1,798 projections through a full 360° rotation of the specimens, depending upon sample width on the instrument detector, with an integration time of 0.5 s per projection, which led to a 10- to 15-min scan time. Large specimens were scanned twice using a multiscan feature to capture the full specimen structure.

Micro-CT data were reconstructed using the datos|x reconstruction software (Waygate Technologies), applying an ROI and inline median filter during the reconstruction of the data. Reconstructed data were imported into Avizo version 2019.3 (Thermo Scientific) for analysis. The structure of each coral specimen was evaluated by segmenting three different phases observed in scans, namely, the dense skeletal phase (bright white structure in Fig. 5a), a lower-density organic phase (intermediate gray values in Fig. 5a), and a phase with trapped air within the structure (dark gray-black space in Fig. 5a). The auto threshold algorithm of Avizo was used for segmentation of the 3 phases (Fig. 5b); this algorithm is based on a factorization method developed by Otsu (86) and determines the point for segmentation between phases in the grayscale histogram (Fig. 5c). To determine the total porosity ($V_{porosity}$; see equation 1a) of specimens, a sample mask was created using the "closing" and "fill holes" operations of Avizo on the segmented skeleton ($V_{skeleton}$; equation 1a) plus organic matter ($V_{organic}$; equation 1a) to produce a solid sample mask ($V_{sample}$; equation 1a) encompassing the boundaries of the sample. The total volume of segmented air ($V_{pores}$; equation 1b) and organic phase ($V_{organic}$; equation 1b) within the sample mask is then taken as a measure of sample porosity (equation 1b). Segmented label volumes were calculated using the "volume fraction" algorithm in Avizo, which also determines the volume fraction of each phase relative to the segmented sample mask. To determine whether the 20-$\mu$m resolution scan sufficiently captured microporosity within the skeleton, the 10-$\mu$m scans were registered to the 20-$\mu$m scans using the "register images" algorithm of Avizo. The same 12 mm by 12 mm by 12 mm ROI was then extracted from the 20-$\mu$m data set, and trends in porosity were compared between the two data sets at different resolutions (Fig. 5d to f).

$$V_{sample} = V_{skeleton} + V_{porosity} \qquad (1a)$$

$$V_{porosity} = V_{organic} + V_{pores} \qquad (1b)$$

Bulk skeletal density (dry weight of the skeleton/$V_{sample}$ [g/cm³]) and porosity ($V_{porosity}/V_{sample} \times 100$ [%]) were the two main parameters we derived from downstream analysis of the skeletal architecture data set. These two parameters are strictly correlated with each other; indeed, as porosity decreases, bulk skeletal density approaches microdensity, and neither can exceed the density of pure aragonite (2.94 mg · mm$^{-3}$ [87]). The complementarity of the two parameters also has implications for the analytical procedure used to measure them, resulting in multicollinearity in downstream analyses. Therefore, we decided to retain only the parameter porosity as representative of skeletal architecture.

**Reproductive mode.** The reproductive mode of each coral species was retrieved from the Coral Trait Database (CTDB) v. 1.1.1 (https://coraltraits.org/) (88), and we divided the species into the following three groups: broadcast spawners (*A. aspera, G. retiformis, G. tenuidens, M. digitata, P. annae, P. australensis, P. daedalea, P. lutea,* and *P. sinensis*), brooders (*I. palifera* and *S. pistillata*), and mixed (*P. damicornis*).

**Statistical analysis.** The significance level for statistical analyses was a $P$ value of $\leq$0.05 and, unless stated otherwise, all analyses were conducted on rarefied amplicon sequence variant (ASV) tables (10,000 sequences per sample). The observed richness and Pielou's evenness ($\alpha$-diversity) of coral skeleton and tissue were computed on unrarefied ASV tables, and differences were tested using Welch's *t* test or the Mann-Whitney *U* test. Differences in community composition ($\beta$-diversity) among coral skeleton, tissue, seawater, and sediment microbiomes were computed using centered log-transformed Euclidean distance matrices of the unrarefied ASV table. Differences among groups were tested using nonparametric multivariate analysis of variance (NPMANOVA) and linear discriminant analysis effect size (LEfSe) (89) and visualized with principal-component analysis (PCA). LEfSe, implemented in *microbiomeMarker* (90), was used to identify differentially abundant ASVs among groups by coupling statistical tests to assess their differences with further tests encoding biological consistency and effect relevance. ASVs with a log(LDA) of >2 (Kruskal-Wallis test: $P < 0.05$) were considered differentially abundant. Z-score-transformed abundance profiles of LEfSe identified ASVs (present in at least 10 samples, inclusive of tissue and skeleton), which were visualized using a heatmap (Fig. 4) via *pheatmap* (91). Associations between the microbiome and predictor variables (host phylogeny, skeletal architecture, and reproductive mode) were assessed via variation partitioning analysis (92) and CCA (93), using Hellinger-transformed, rarefied ASV tables. CCA is a multivariate method that can disentangle patterns or changes in biological communities and identify potential associations between explanatory variables and ASVs (based on their coordinates' similarity on the CCA plots). Both analyses (variation partitioning and CCA) were performed on the top 200 most abundant ASVs of the data set, both to reduce its dimension and because rare species may have an unduly large influence on these types of analysis (94). Statistical collinearity of the predictor variables was assessed using RStudio version 1.2.5033 via *olsrr* (95), and colinear variables were removed from the analysis. We calculated the 30%, 50%, and 60% core and rare microbiomes as ASVs that were and were not consistently present at these three thresholds, respectively. All analyses were conducted using RStudio version 1.2.5033 and the packages *agricolae*

(96), *ampvis2* (97), *ape* (98), *decontam* (76), *dplyr* (99), *ggfortify* (100), *ggplot2* (101), *ggvegan* (102), *microbiome* (103), *microbiomeMarker* (90), *pheatmap* v1.0.12 (91), *phyloseq* (104), *rgr* (105), and *vegan* (106).

**Ethics approval.** All of the coral samples were collected under permit G19/41658.1 issued by the Great Barrier Reef Marine Park Authority.

**Data availability.** Sequence data determined in this study are available at NCBI under SRA accession number PRJNA719930. Supplemental tables, data analysis workflow, raw ASV tables, and plots are available at https://melbourne.figshare.com/projects/Host_traits_and_phylogeny_contribute_to_shaping_coral -bacterial_symbioses/118896.

## SUPPLEMENTAL MATERIAL

Supplemental material is available online only.
**FIG S1**, TIF file, 2.2 MB.
**FIG S2**, TIF file, 1.7 MB.
**FIG S3**, TIF file, 2 MB.
**FIG S4**, TIF file, 2.3 MB.
**TABLE S1**, XLSX file, 8.6 MB.
**TABLE S2**, XLSX file, 0.02 MB.
**TABLE S3**, XLSX file, 0.02 MB.
**TABLE S4**, XLSX file, 0.01 MB.
**TABLE S5**, XLSX file, 1 MB.
**TABLE S6**, XLSX file, 0.01 MB.

## ACKNOWLEDGMENTS

We thank the staff at Heron Island Research Station for assisting and Stephen Wilcox from the Walter and Eliza Hall Institute for supervising the molecular analysis and facilitating sequencing. We acknowledge the Melbourne TrACEES Platform (Trace Analysis for Chemical, Earth and Environmental Sciences) for access to the micro-CT scanner. K.T. is thankful to Sen-Lin Tang from the Biodiversity Research Center, Academia Sinica, Taiwan, for hosting him in his laboratory.

We acknowledge funding from the Australian Research Council (grant DP200101613 to H.V. and L.L.B.), from a National Health and Medical Research Council (NHMRC) Career Development fellowship (GNT1159458 to K.-A.L.C.), rom the Environmental Microbiology Research Initiative at the University of Melbourne (to F.R.), from the Native Australian Animal Trust (to F.R.), and from the Holsworth Wildlife Research Endowment (to F.R.).

F.R. and H.V. contributed to the conceptual development of the manuscript. F.R. conducted the experiments. F.R., K.T., J.B., and H.V. conducted the data analysis. All authors contributed to the final edited version of th manuscript.

On behalf of all authors, the corresponding author states that there is no conflict of interest.

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
