## [Reviewer comments · mSystems]

Host traits and phylogeny contribute to shaping coral-bacterial symbioses

Francesco Ricci, Kshitij Tandon, Jay Black, Kim-Anh Lê Cao, Linda Blackall, and Heroen Verbruggen

Corresponding Author(s): Francesco Ricci, University of Melbourne

Review Timeline:

Submission Date:	January 17, 2022
Editorial Decision:	February 1, 2022
Revision Received:	February 2, 2022
Editorial Decision:	February 12, 2022
Revision Received:	February 13, 2022
Accepted:	February 14, 2022

Editor: Jean-Baptiste Raina

Reviewer(s): The reviewers have opted to remain anonymous.

Transaction Report:

DOI: <https://doi.org/10.1128/msystems.00044-22>

February 1, 2022

Mr. Francesco Ricci
University of Melbourne
Melbourne
Australia

Re: mSystems00044-22 (Host traits and phylogeny contribute to shaping coral-bacterial symbioses)

Dear Francesco,

Thank you for submitting your manuscript to mSystems. We have completed our review and I am pleased to inform you that, in principle, we expect to accept it for publication in mSystems. However, acceptance will not be final until you have adequately addressed the reviewer comments.

Below you will find instructions from the mSystems editorial office and comments generated during the review.

Preparing Revision Guidelines

Sincerely,

Jean-Baptiste Raina

Editor, mSystems

Journals Department
Reviewer comments:

Reviewer #1 (Comments for the Author):

I have already served as reviewer 1 in the previous submission of this manuscript. Overall, the authors have clearly put effort into the revision. The manuscript has improved, however there are still a few things that require clarification.

The main thing that still bothers me is the high number of PCR cycles (24 cycles in the indexing PCR step, totaling 50 cycles in total), which I find problematic. There are multiple publications targeting 16S with PCR using DNA isolated from coral skeletons that do not exceed 35 cycles (including the indexing PCR), and while I agree that such DNA is not the easiest of templates, it is doable. The authors of course cannot change this aspect of their work at this stage, but the limitations of excessive PCR cycling should at least be acknowledged in the methods (accumulation of non-specific random-length fragments - which will ultimately reduce the yield of specific product; see Bell & deMarini 1991 *Nucleic Acids Research* for reference).

In addition, there is still a degree of typos / minor grammatical errors throughout the manuscript, in particular the marked up sections. I would recommend the authors have a native speaker / language editing service take a look at the manuscript, or run it through Grammarly (or similar).

More specific comments

Line 27 - perhaps replace 'hypothesize' with 'conclude' ?

Line 31 - apologies I did not pick up on this wording here before - I suggest replacing 'we found that a large fraction of bacteria can colonize both anatomical compartments' with 'we found that a large overlapping fraction of bacterial sequences were recovered from both anatomical compartments' - with this study design, *colonization cannot be shown*, only inferred from sequencing data, and discussed.

Lines 44-45: 'we show that the partnerships with bacterial symbionts critical to coral health' - I know this is a suggestion by the other reviewer, but I would strongly suggest toning this down (as the authors have done in the discussion section, where they keep referring to 'putative important bacteria'). There is a lot of hypothesizing and musing about the importance of coral bacteria but a lot of it is inferred from relative abundance data in sequencing datasets or from genomic data. There is in my opinion still too little functional work to support this statement beyond very few functional groups. Perhaps a compromise for the authors could be 'we show that the association with bacterial symbionts thought to be critical to coral health (..)'

Line 220: context unclear - what is the number range '89-290' here - are these numbers of ASVs?

Line 237 - grammar - 'the skeleton transports and redistributes' (rather than 'the skeleton transport and redistribute')

Line 271: wording - suggest replacing with 'including nutrient provision and support of host homeostasis'

Lines 279-281: this sentence is in part an exact repetition of 269/270

Line 331-340: the authors have gone a bit overboard with my previous suggestion to include evenness for the "'extreme' cases (like *G. retiformes* or *P. australensis*)" - With 'adding some numbers' I meant adding means for these two species and their two compartments - it is excessive to provide minimum, mean, and maximum values for evenness. Apologies for the confusion.

Lines 377 and following, Lines 390 and following: 'Bacteria in the genus *Paramaledivibacter* accounted for the 2.92% of the dataset ASVs' / '*Roseospira* (...), but ASVs affiliated to these bacteria constituted the 0.29% of the datasets' - I would recommend the authors seek language editing services to iron out the last of the wrinkles in their writing.

Line 409: suggest replacing 'were associate with a range of microbial associates' with 'were associated with a range of microbial partners', or similar (get rid of repetition and fix tense)

Line 449: corals were collected using a sterile hammer and chisel - were hammer and chisel cleaned between individual fragments? Otherwise they are not really going to stay clean for long.

Lines 464-467: kindly clarify that the 'control sample' and 'blanks' are mock samples to account for kit contamination. Importantly: were both the 'control samples' and the 'blank extractions' subjected to PCR and sequencing (only controls mentioned in line 481)?

Reviewer #2 (Comments for the Author):

Even though the authors addressed most of the comments made by both reviewers, I believe the manuscript would still benefit from revising the results and discussion section, which would mostly require rephrasing to improve conciseness and clarity.

Minor concerns:

Line 60: Perhaps the authors could replace by our "understanding" of the key mechanisms driving host-symbiont assemblages is limited...

Line 120: I would suggest mentioning the link of supplementary data at the end of the manuscript in the Supplementary material section, and this for all files. This section is already a bit lengthy, and these links add to this length. Also, when clicking on the link in the manuscript I get an error "page cannot be found", but the link at the end of the manuscript is working. Would it be easier for the readers to have access to the supplementary materials directly from the home page of the manuscript?

Line 123: Was the proportion of unclassified read high? Could you please precise.

Lines 125-129: I am not convinced that the statistics for the number of read per sample is necessary, but perhaps this was asked by another reviewer.

Line 130: May I suggest replacing by "but differ in their composition" rather than their relative abundances?

Lines 135-137: I feel this general statement should be placed further down as the authors investigated the coral microbiome of visibly healthy colonies.

Line 154: If I am correct, Pollock et al. found differences in α -diversities. If so, please precise for clarity.

Line 183: Since the authors mentioned the "positive correlation" I suggest replacing by: ASVs belonging to this genus are more abundant in corals of the robust clade, the families Poritidae and Pocilloporidae (positive correlations with PV1, PV2, PV3, respectively). Such rephrasing throughout the Results and discussion section would help the readers and improve conciseness and clarity of the manuscript.

Lines 190-199: I would appreciate a hypothesis explaining *Alteromonas* being more abundant in corals of the robust clade and *Pocilloporidae*, while *Pseudoalteromonas* being more abundant in *Poritidae* corals.

Lines 233-235: Maybe I misunderstood, but how is the variation partitioning analysis show that the characteristic skeletal architecture of each coral species correlated with the composition of tissue and skeletal microbiomes? Would it be more accurate to say: The characteristic skeletal architecture of each coral species explained a portion of the microbiome variation (3% for both the tissue and skeleton).

Lines 266-269: To which figure or table the authors are referring for the higher abundances of *Acinetobacter* spp., *Bacillus* spp., *Caulobacteriales*, *Cryomorphaceae*, *Rhizobiales*, and *Rhodobacteriales*?

Lines 276-277: Our data show that tissue and skeletal bacteria correlated with the reproductive mode variables (broadcast spawners, brooders and mixed mode; Fig. 3a and 3d). Should it refer to Fig. 3b and 3d instead?

Line 279-281: This is a repetition of lines 269-271.

Line 302: I suggest replacing by "i.e. five ASVs occurring in at least 50% of tissue samples and one for the skeleton, and one single ASV occurring in at least 60% of both tissue and skeleton samples, while we could not...."

Table S1: It would be informative to know the counts and taxonomic annotation of each ASV.

Table S2: It would also be informative to know the relative proportion of each ASV.

Table S3: Are the numbers referring to relative abundances, as a proportion or percentages? Please precise in the table title

Table S4: It would be informative to know the relative abundance and counts for core members.

Table S5: Are the numbers referring to the number of reads? Please precise in the table title?

Table S5: Are the numbers referring to a percentage? Please precise in the table title.

Reviewer #1 (Comments for the Author):

I have already served as reviewer 1 in the previous submission of this manuscript. Overall, the authors have clearly put effort into the revision. The manuscript has improved, however there are still a few things that require clarification.

The main thing that still bothers me is the high number of PCR cycles (24 cycles in the indexing PCR step, totaling 50 cycles in total), which I find problematic. There are multiple publications targeting 16S with PCR using DNA isolated from coral skeletons that do not exceed 35 cycles (including the indexing PCR), and while I agree that such DNA is not the easiest of templates, it is doable. The authors of course cannot change this aspect of their work at this stage, but the limitations of excessive PCR cycling should at least be acknowledged in the methods (accumulation of non-specific random-length fragments - which will ultimately reduce the yield of specific product; see Bell & deMarini 1991 Nucleic Acids Research for reference).

Agreed. Pag 18 lines 433-436. We've added the following "We note that the total number of cycles (49) in the 2-step PCR amplification could have led to accumulation of non-specific products, however, the quality of the PCR products was tested on a TapeStation (model 4200) at the Walter and Eliza Hall Institute of Medical Research."

In addition, there is still a degree of typos / minor grammatical errors throughout the manuscript, in particular the marked up sections. I would recommend the authors have a native speaker / language editing service take a look at the manuscript, or run it through Grammarly (or similar).

Agreed. The text has been proof-read by a native English speaker.

More specific comments

Line 27 - perhaps replace 'hypothesize' with 'conclude' ?

Agreed

Line 31 - apologies I did not pick up on this wording here before - I suggest replacing 'we found that a large fraction of bacteria can colonize both anatomical compartments' with 'we found that a large overlapping fraction of bacterial sequences were recovered from both anatomical compartments' - with this study design, *colonization cannot be shown*, only inferred from sequencing data, and discussed.

Agreed

Lines 44-45: 'we show that the partnerships with bacterial symbionts critical to coral health' - I know this is a suggestion by the other reviewer, but I would strongly suggest toning this down (as the authors have done in the discussion section, where they keep referring to 'putative important bacteria'). There is a lot of hypothesizing and musing about the importance of coral bacteria but a lot of it is inferred from

relative abundance data in sequencing datasets or from genomic data. There is in my opinion still too little functional work to support this statement beyond very few functional groups. Perhaps a compromise for the authors could be 'we show that the association with bacterial symbionts thought to be critical to coral health (..)'

Agreed

Line 220: context unclear - what is the number range '89-290' here - are these numbers of ASVs?

Agreed. Numbers have been removed.

Line 237 - grammar - 'the skeleton transports and redistributes' (rather than 'the skeleton transport and redistribute')

Agreed.

Line 271: wording - suggest replacing with 'including nutrient provision and support of host homeostasis'

Agreed.

Lines 279-281: this sentence is in part an exact repetition of 269/270

Agreed. Text has been changed to "These bacteria are all known for their putative beneficial roles and could help the host by being involved in processes such as nutrient cycling and support of homeostasis"

Line 331-340: the authors have gone a bit overboard with my previous suggestion to include evenness for the "'extreme' cases (like *G. retiformes* or *P. australensis*)" - With 'adding some numbers' I meant adding means for these two species and their two compartments - it is excessive to provide minimum, mean, and maximum values for evenness. Apologies for the confusion.

Agreed. Only mean values are now reported.

Lines 377 and following, Lines 390 and following: 'Bacteria in the genus *Paramaledivibacter* accounted for the 2.92% of the dataset ASVs' / '*Roseospira* (...), but ASVs affiliated to these bacteria constituted the 0.29% of the datasets' - I would recommend the authors seek language editing services to iron out the last of the wrinkles in their writing.

Agreed. Pag 14 lines 324 and 332, Pag 15 line 344.

Line 409: suggest replacing 'were associate with a range of microbial associates' with 'were associated with a range of microbial partners', or similar (get rid of repetition and fix tense)

Agreed.

Line 449: corals were collected using a sterile hammer and chisel - were hammer and chisel cleaned between individual fragments? Otherwise they are not really going to stay clean for long.

Agreed. Text has been changed to “Corals were collected using a set of sterile hammers and chisels”

Liens 464-467: kindly clarify that the 'control sample' and 'blanks' are mock samples to account for kit contamination. Importantly: were both the 'control samples' and the 'blank extractions' subjected to PCR and sequencing (only controls mentioned in line 481)?

Agreed. Pag 18 Line 416 and 431.

Reviewer #2 (Comments for the Author):

Even though the authors addressed most of the comments made by both reviewers, I believe the manuscript would still benefit from revising the results and discussion section, which would mostly require rephrasing to improve conciseness and clarity.

Agreed. Trying to meet Reviewer 2 request we made some further modifications to the results and discussion and the text has been proofread by a native english speaker author.

Minor concerns:

Line 60: Perhaps the authors could replace by our "understanding" of the key mechanisms driving host-symbiont assemblages is limited...

Agreed

Line 120: I would suggest mentioning the link of supplementary data at the end of the manuscript in the Supplementary material section, and this for all files. This section is already a bit lengthy, and these links add to this length. Also, when clicking on the link in the manuscript I get an error "page cannot be found", but the link at the end of the manuscript is working. Would it be easier for the readers to have access to the supplementary materials directly from the home page of the manuscript?

Agreed. The links throughout the text have been removed.

Line 123: Was the proportion of unclassified read high? Could you please precise.

Agreed. Pag 5 lines 120-121. We added the text “The dataset did not include unclassified reads (Table S1)”

Lines 125-129: I am not convinced that the statistics for the number of read per sample is necessary, but perhaps this was asked by another reviewer.

We will keep these statistics for clarity.

Line 130: May I suggest replacing by "but differ in their composition" rather than their relative abundances?

Disagree. The composition is actually not different, what differs is the relative number of reads.

Lines 135-137: I feel this general statement should be placed further down as the authors investigated the coral microbiome of visibly healthy colonies.

Disagree. This is the core of the reservoir hypothesis formulated by Marcelino et al.

Line 154: If I am correct, Pollock et al. found differences in α -diversities. If so, please precise for clarity.

Agreed. Page 6 line 144-145. We added the text "This conflicts with the findings of Pollock et al. (6) that found differences in the α -diversities of the two anatomical compartments"

Line 183: Since the authors mentioned the "positive correlation" I suggest replacing by: ASVs belonging to this genus are more abundant in corals of the robust clade, the families Poritidae and Pocilloporidae (positive correlations with PV1, PV2, PV3, respectively). Such rephrasing throughout the Results and discussion section would help the readers and improve conciseness and clarity of the manuscript.

Agreed. We applied Reviewer 2 suggestion where appropriate. E.g. Pag 7 line 169.

Lines 190-199: I would appreciate a hypothesis explaining *Alteromonas* being more abundant in corals of the robust clade and Pocilloporidae, while *Pseudoalteromonas* being more abundant in Poritidae corals.

Agreed. Pag 8 lines 183-186. We hypothesize that the absence of correlation between *Alteromonas* and Poritidae, which suggest low relative abundance of these bacteria in this coral lineage, could be driven by competition for similar resources between these and other bacteria.

Lines 233-235: Maybe I misunderstood, but how is the variation partitioning analysis show that the characteristic skeletal architecture of each coral species correlated with the composition of tissue and skeletal microbiomes? Would it be more accurate to say: The characteristic skeletal architecture of each coral species explained a portion of the microbiome variation (3% for both the tissue and skeleton).

Agreed. Page 9 line 216-18. The text has been changed accordingly.

Lines 266-269: To which figure or table the authors are referring for the higher abundances of *Acinetobacter* spp., *Bacillus* spp., Caulobacterales, Cryomorpaceae, Rhizobiales, and Rhodobacterales?

Agreed. These taxa have now been included in Figure 3b.

Lines 276-277: Our data show that tissue and skeletal bacteria correlated with the

reproductive mode variables (broadcast spawners, brooders and mixed mode; Fig. 3a and 3d). Should it refer to Fig. 3b and 3d instead?

Agreed. Pag 10 line 253. Text has been changed to Fig. 3b and 3c.

Line 279-281: This is a repetition of lines 269-271.

Agreed. Text has been changed to "These bacteria are all known for their putative beneficial roles and could help the host by being involved in processes such as nutrient cycling and support of homeostasis"

Line 302: I suggest replacing by "i.e. five ASVs occurring in at least 50% of tissue samples and one for the skeleton, and one single ASV occurring in at least 60% of both tissue and skeleton samples, while we could not...."

Agreed. Pag 12 lines 274-278. Text has been changed accordingly.

Table S1: It would be informative to know the counts and taxonomic annotation of each ASV.

Agreed. This information is already enclosed in Table S1.

Table S2: It would also be informative to know the relative proportion of each ASV.

The aim of this Table is showing which taxa were identified as differentially abundant between tissue and skeleton, the information requested by Review 2 is not in line with the purpose of this Table. Furthermore, the information requested by Reviewer 2 is also present in Table S1.

Table S3: Are the numbers referring to relative abundances, as a proportion or percentages? Please precise in the table title

Agreed. Text has been added to Table S3 caption.

Table S4: It would be informative to know the relative abundance and counts for core members.

Inserting that information would deviate from the purpose of the table making its interpretation more complicated than necessary. Furthermore, the information requested by Reviewer 2 is also present in Table 5.

Table S5: Are the numbers referring to the number of reads? Please precise in the table title?

Agreed. Yes, this information was already enclosed in Table S5 caption.

Table S6: Are the numbers referring to a percentage? Please precise in the table title.

Agreed. Yes these are relative abundance. This information was already enclosed in Table S6 caption.

February 12, 2022

Mx. Francesco Ricci
University of Melbourne
Melbourne
Australia

Re: mSystems00044-22R1 (Host traits and phylogeny contribute to shaping coral-bacterial symbioses)

Dear Francesco,

I am satisfied with your response to the reviewers' comments. Given that both reviewers have previously commented on some minor grammatical issues, I have made some edits and suggestions attached. You do not have to implement them all, but I do believe that these edits will further improve the readability of the text. Please go through my edits (no need to do a point-by-point response) and resubmit, I will then formally accept it for publication.

Sincerely,

Jean-Baptiste Raina

Editor, mSystems

Journals Department
Reviewer comments:

Reviewer #1 (Comments for the Author):

I have already served as reviewer 1 in the previous submission of this manuscript. Overall, the authors have clearly put effort into the revision. The manuscript has improved, however there are still a few things that require clarification.

The main thing that still bothers me is the high number of PCR cycles (24 cycles in the indexing PCR step, totaling 50 cycles in total), which I find problematic. There are multiple publications targeting 16S with PCR using DNA isolated from coral skeletons that do not exceed 35 cycles (including the indexing PCR), and while I agree that such DNA is not the easiest of templates, it is doable. The authors of course cannot change this aspect of their work at this stage, but the limitations of excessive PCR cycling should at least be acknowledged in the methods (accumulation of non-specific random-length fragments - which will ultimately reduce the yield of specific product; see Bell & deMarini 1991 Nucleic Acids Research for reference).

Agreed. Pag 18 lines 433-436. We've added the following "We note that the total number of cycles (49) in the 2-step PCR amplification could have led to accumulation of non-specific products, however, the quality of the PCR products was tested on a TapeStation (model 4200) at the Walter and Eliza Hall Institute of Medical Research."

In addition, there is still a degree of typos / minor grammatical errors throughout the manuscript, in particular the marked up sections. I would recommend the authors have a native speaker / language editing service take a look at the manuscript, or run it through Grammarly (or similar).

Agreed. The text has been proof-read by a native English speaker.

More specific comments

Line 27 - perhaps replace 'hypothesize' with 'conclude' ?

Agreed

Line 31 - apologies I did not pick up on this wording here before - I suggest replacing 'we found that a large fraction of bacteria can colonize both anatomical compartments' with 'we found that a large overlapping fraction of bacterial sequences were recovered from both anatomical compartments' - with this study design, *colonization cannot be shown*, only inferred from sequencing data, and discussed.

Agreed

Lines 44-45: 'we show that the partnerships with bacterial symbionts critical to coral health' - I know this is a suggestion by the other reviewer, but I would strongly suggest toning this down (as the authors have done in the discussion section, where they keep referring to 'putative important bacteria'). There is a lot of hypothesizing and musing about the importance of coral bacteria but a lot of it is inferred from

relative abundance data in sequencing datasets or from genomic data. There is in my opinion still too little functional work to support this statement beyond very few functional groups. Perhaps a compromise for the authors could be 'we show that the association with bacterial symbionts thought to be critical to coral health (..)'

Agreed

Line 220: context unclear - what is the number range '89-290' here - are these numbers of ASVs?

Agreed. Numbers have been removed.

Line 237 - grammar - 'the skeleton transports and redistributes' (rather than 'the skeleton transport and redistribute')

Agreed.

Line 271: wording - suggest replacing with 'including nutrient provision and support of host homeostasis'

Agreed.

Lines 279-281: this sentence is in part an exact repetition of 269/270

Agreed. Text has been changed to "These bacteria are all known for their putative beneficial roles and could help the host by being involved in processes such as nutrient cycling and support of homeostasis"

Line 331-340: the authors have gone a bit overboard with my previous suggestion to include evenness for the "'extreme' cases (like *G. retiformes* or *P. australensis*)" - With 'adding some numbers' I meant adding means for these two species and their two compartments - it is excessive to provide minimum, mean, and maximum values for evenness. Apologies for the confusion.

Agreed. Only mean values are now reported.

Lines 377 and following, Lines 390 and following: 'Bacteria in the genus *Paramaledivibacter* accounted for the 2.92% of the dataset ASVs' / '*Roseospira* (...), but ASVs affiliated to these bacteria constituted the 0.29% of the datasets' - I would recommend the authors seek language editing services to iron out the last of the wrinkles in their writing.

Agreed. Pag 14 lines 324 and 332, Pag 15 line 344.

Line 409: suggest replacing 'were associate with a range of microbial associates' with 'were associated with a range of microbial partners', or similar (get rid of repetition and fix tense)

Agreed.

Line 449: corals were collected using a sterile hammer and chisel - were hammer and chisel cleaned between individual fragments? Otherwise they are not really going to stay clean for long.

Agreed. Text has been changed to “Corals were collected using a set of sterile hammers and chisels”

Liens 464-467: kindly clarify that the 'control sample' and 'blanks' are mock samples to account for kit contamination. Importantly: were both the 'control samples' and the 'blank extractions' subjected to PCR and sequencing (only controls mentioned in line 481)?

Agreed. Pag 18 Line 416 and 431.

Reviewer #2 (Comments for the Author):

Even though the authors addressed most of the comments made by both reviewers, I believe the manuscript would still benefit from revising the results and discussion section, which would mostly require rephrasing to improve conciseness and clarity.

Agreed. Trying to meet Reviewer 2 request we made some further modifications to the results and discussion and the text has been proofread by a native english speaker author.

Minor concerns:

Line 60: Perhaps the authors could replace by our "understanding" of the key mechanisms driving host-symbiont assemblages is limited...

Agreed

Line 120: I would suggest mentioning the link of supplementary data at the end of the manuscript in the Supplementary material section, and this for all files. This section is already a bit lengthy, and these links add to this length. Also, when clicking on the link in the manuscript I get an error "page cannot be found", but the link at the end of the manuscript is working. Would it be easier for the readers to have access to the supplementary materials directly from the home page of the manuscript?

Agreed. The links throughout the text have been removed.

Line 123: Was the proportion of unclassified read high? Could you please precise.

Agreed. Pag 5 lines 120-121. We added the text “The dataset did not include unclassified reads (Table S1)”

Lines 125-129: I am not convinced that the statistics for the number of read per sample is necessary, but perhaps this was asked by another reviewer.

We will keep these statistics for clarity.

Line 130: May I suggest replacing by "but differ in their composition" rather than their relative abundances?

Disagree. The composition is actually not different, what differs is the relative number of reads.

Lines 135-137: I feel this general statement should be placed further down as the authors investigated the coral microbiome of visibly healthy colonies.

Disagree. This is the core of the reservoir hypothesis formulated by Marcelino et al.

Line 154: If I am correct, Pollock et al. found differences in α -diversities. If so, please precise for clarity.

Agreed. Page 6 line 144-145. We added the text "This conflicts with the findings of Pollock et al. (6) that found differences in the α -diversities of the two anatomical compartments"

Line 183: Since the authors mentioned the "positive correlation" I suggest replacing by: ASVs belonging to this genus are more abundant in corals of the robust clade, the families Poritidae and Pocilloporidae (positive correlations with PV1, PV2, PV3, respectively). Such rephrasing throughout the Results and discussion section would help the readers and improve conciseness and clarity of the manuscript.

Agreed. We applied Reviewer 2 suggestion where appropriate. E.g. Pag 7 line 169.

Lines 190-199: I would appreciate a hypothesis explaining *Alteromonas* being more abundant in corals of the robust clade and Pocilloporidae, while *Pseudoalteromonas* being more abundant in Poritidae corals.

Agreed. Pag 8 lines 183-186. We hypothesize that the absence of correlation between *Alteromonas* and Poritidae, which suggest low relative abundance of these bacteria in this coral lineage, could be driven by competition for similar resources between these and other bacteria.

Lines 233-235: Maybe I misunderstood, but how is the variation partitioning analysis show that the characteristic skeletal architecture of each coral species correlated with the composition of tissue and skeletal microbiomes? Would it be more accurate to say: The characteristic skeletal architecture of each coral species explained a portion of the microbiome variation (3% for both the tissue and skeleton).

Agreed. Page 9 line 216-18. The text has been changed accordingly.

Lines 266-269: To which figure or table the authors are referring for the higher abundances of *Acinetobacter* spp., *Bacillus* spp., Caulobacterales, Cryomorpaceae, Rhizobiales, and Rhodobacterales?

Agreed. These taxa have now been included in Figure 3b.

Lines 276-277: Our data show that tissue and skeletal bacteria correlated with the

reproductive mode variables (broadcast spawners, brooders and mixed mode; Fig. 3a and 3d). Should it refer to Fig. 3b and 3d instead?

Agreed. Pag 10 line 253. Text has been changed to Fig. 3b and 3c.

Line 279-281: This is a repetition of lines 269-271.

Agreed. Text has been changed to "These bacteria are all known for their putative beneficial roles and could help the host by being involved in processes such as nutrient cycling and support of homeostasis"

Line 302: I suggest replacing by "i.e. five ASVs occurring in at least 50% of tissue samples and one for the skeleton, and one single ASV occurring in at least 60% of both tissue and skeleton samples, while we could not...."

Agreed. Pag 12 lines 274-278. Text has been changed accordingly.

Table S1: It would be informative to know the counts and taxonomic annotation of each ASV.

Agreed. This information is already enclosed in Table S1.

Table S2: It would also be informative to know the relative proportion of each ASV.

The aim of this Table is showing which taxa were identified as differentially abundant between tissue and skeleton, the information requested by Review 2 is not in line with the purpose of this Table. Furthermore, the information requested by Reviewer 2 is also present in Table S1.

Table S3: Are the numbers referring to relative abundances, as a proportion or percentages? Please precise in the table title

Agreed. Text has been added to Table S3 caption.

Table S4: It would be informative to know the relative abundance and counts for core members.

Inserting that information would deviate from the purpose of the table making its interpretation more complicated than necessary. Furthermore, the information requested by Reviewer 2 is also present in Table 5.

Table S5: Are the numbers referring to the number of reads? Please precise in the table title?

Agreed. Yes, this information was already enclosed in Table S5 caption.

Table S6: Are the numbers referring to a percentage? Please precise in the table title.

Agreed. Yes these are relative abundance. This information was already enclosed in Table S6 caption.

February 14, 2022

Mx. Francesco Ricci
University of Melbourne
Melbourne
Australia

Re: mSystems00044-22R2 (Host traits and phylogeny contribute to shaping coral-bacterial symbioses)

Dear Francesco:

Your manuscript has been accepted, and I am forwarding it to the ASM Journals Department for publication. For your reference, ASM Journals' address is given below. Before it can be scheduled for publication, your manuscript will be checked by the mSystems production staff to make sure that all elements meet the technical requirements for publication. They will contact you if anything needs to be revised before copyediting and production can begin. Otherwise, you will be notified when your proofs are ready to be viewed.

Publication Fees:

We recognize that the video files can become quite large, and so to avoid quality loss ASM suggests sending the video file via <https://www.wetransfer.com/>. When you have a final version of the video and the still ready to share, please send it to mSystems staff at mssystemsjournal@msubmit.net.

For mSystems research articles, if you would like to submit an image for consideration as the Featured Image for an issue, please contact mSystems staff at mssystemsjournal@msubmit.net.

Sincerely,

Jean-Baptiste Raina

Editor, mSystems

Journals Department
Table S4: Accept

Table S1: Accept

Table S3: Accept

Table S6: Accept

Table S5: Accept

Fig. S4: Accept

Fig. S1: Accept

Fig. S3: Accept

Fig. S2: Accept

Table S2: Accept